# Characterization of depth perception information inferred from neuronal activity in primary visual cortex

Nuo Dong, Yuping Tan, Yuyuan Wang, Yumin Chen, Haibing Xu[ID]*

Department of Neurobiology, School of Basic Medical Sciences, Southern Medical University, Guangdong-Hong Kong-Macao Greater Bay Area Center for Brain Science and Brain-Inspired Intelligence, Guangzhou, Guangdong, China

* haibingxu@smu.edu.cn

## Abstract

Depth perception is crucial for spatial awareness, enabling animals to interpret three-dimensional environments. Although the primary visual cortex (V1) is known to process depth information, the specific contributions of V1 neurons to various aspects of depth perception remain underexplored. In this study, we investigated how V1 neurons engage in both passive and active depth-related tasks, examining whether distinct neuronal populations support different aspects of depth processing. Using in vivo calcium imaging in freely moving mice, we observed that specific groups of V1 neurons are selectively active in passive (visual cliff) and active (depth discrimination) tasks, suggesting functional segregation within V1. Additionally, neurons in the primary visual cortex prefer encoding objective positions rather than egocentric distances in non-depth-based tasks. Moreover, egocentric distance discrimination, as reflected by the primary visual cortex, appears to be more prospective. These findings provide insight into V1's versatility, highlighting its potential role in spatial navigation and decision-making.

## Introduction

Depth perception is essential for understanding and navigating a three-dimensional world [1], relying on various monocular (e.g., retinal image size, motion parallax) and binocular cues (e.g., disparity, convergence) to estimate distances accurately [2]. While early research emphasized static cues such as occlusion, shading, and perspective, recent studies highlight the significant role of dynamic cues like motion parallax, especially when observer and object are in relative motion, for accurately perceiving depth [3,4]. This multi-faceted perception is orchestrated by the visual cortex, which processes a range of visual cues associated with depth and spatial information [5–8].

**Data availability statement:** The datasets and code necessary to replicate the analyses for this project are available at OSF (https://osf.io/ec4fd/).

**Funding:** STI2030-Major Projects, 2022ZD0205000, Prof. Haibing Xu National Natural Science Foundation of China, 32171038, Prof. Haibing Xu National Natural Science Foundation of China, W243058, Prof. Haibing Xu Basic and Applied Basic Research Foundation of Guangdong Province, 2022A515011896, Prof. Haibing Xu Basic and Applied Basic Research Foundation of Guangdong Province, 2024A1515011033, Prof. Haibing Xu.

**Competing interests:** The authors have declared that no competing interests exist.

Originally, vision was only considered as one of the important drivers in spatial navigation [9–11]. However, the primary visual cortex (V1), the first stage of higher-order processing in the visual pathway, supports numerous cognitive functions and encodes spatially relevant cues within its neurons [12–16], suggests a more dynamic involvement. Notably, location-specific neurons in V1 exhibit place cell-like activity, firing at specific locations within an environment, similar to the hippocampus [14,16]. This expanding view positions V1 as a contributor to spatial cognition and depth processing, potentially integrating these functions through interactions with systems for memory, decision-making, and conscious processing [17–21].

Given V1's sophisticated encoding capabilities [22–24], it raises the question: how does V1 respond to various depth cues under different behavioral contexts? Current studies suggest that depth-related neural responses in V1 may represent distinct depth-related stimuli [22,24]. To investigate this duality, we applied the visual cliff paradigm, a well-established method to elicit passive depth responses in animals, focusing on neuronal populations sensitive to depth stimuli [25,26]. Furthermore, considering the importance of V1's potential in depth-based decision-making [27,28], we developed a novel depth-discrimination task aimed at identifying neurons actively engaged in depth-based judgment. Our approach combined in vivo single-cell recordings during both passive and active tasks to discern potential functional segregation within V1 in response to depth cues.

To further clarify the functional dissociation between neuronal populations involved in passive versus active depth perception, we introduced a linear track to examine the specific activity patterns of these two populations. This study provides insight into how V1 neuronal populations differentially engage in depth perception across passive and active behavioral contexts, suggesting distinct intra-V1 processing pathways that contribute to an animal's depth perception and spatial orientation. Moreover, these results propose that V1 neurons not only passively receive depth information but also actively participate in higher-order cognitive processes related to spatial navigation and memory encoding, reflecting a previously underappreciated versatility within the primary visual cortex.

## Materials and methods

### Animals

Male wild-type C57BL/6 mice, aged 5–6 weeks, obtained from the Southern Medical University Animal Center (Guangzhou, China) and Guangdong Medical Laboratory Animal Center (Guangzhou, China), were group-housed until lens implantation. The mice were maintained in a controlled environment with a reverse 12-hour light/dark cycle, regular bedding replacement, temperature at 18–22°C, and the humidity at 50–60%, having ad libitum access to food and water.

All of the experiments were performed in compliance with the Regulations for the Administration of Affairs Concerning Experimental Animals (China) and were approved by the Southern Medical University Animal Ethics Committee (approval No. L2022096) regarding the care of laboratory animals.

## Surgical procedures

Mice were acclimated to the experimental environment before the procedure. Mice were anesthetized with 1.25% tribro-moethanol at 0.2 ml per 10 g of body weight. The pAAV-hSyn-GCaMp6f-WPRE (OBiO) virus was injected at a rate of 10nL/min at AP: −3.65 mm, ML: −2.25 mm, and DV: −1.00 mm, 2 weeks prior to lens implanting [29].

A 1-mm-diameter bone flap centered on the injection site was removed using a cranial drill. Cranial screws were evenly distributed and affixed around the cranial window. A 1-mm-diameter lens (GoPhoton) held by a vacuum system was zeroed at the fontanel and positioned to the appropriate z-axis depth. The lens was fixed to the skull with dental adhesive and topped with non-transparent silicone for protection. One week after surgery, mice fully recovered and had a baseplate implanted.

## Experimental apparatus and maze design

The visual cliff experiment was conducted in a 60 cm × 60 cm open field with 20cm-high walls. Half of the open field with a transparent bottom panel was placed on a 90 cm high platform, with the other half suspended. Depth was adjusted by adding panels between the bottom panel and the ground. The cliff planes and sagging surface were covered with a 2.5 cm x 2.5 cm checkerboard pattern. The center region was defined as a concentric 30 cm square. Each experiment consisted of two 15-minutes sessions at depths of 30 cm and 90 cm respectively, starting with the mice placed in the center of the open field. Between sessions, mice were briefly removed for depth adjustment and 75% alcohol maze cleaning.

We designed a radial variable-length-arm maze (VAM). The starting area was a hexagonal prism, supporting up to six arms radiating outward. The starting platform was elevated 20 cm, with arms 15 cm high, 7 cm wide, and adjustable up to 120 cm in length. For the experiment, two arms were set to lengths of 20 cm and 40 cm randomly switched, refer to sequences generated by Python random module. A 3D-printed food bowl was placed at the end of each arm. The maze doors were controlled by air pump connected to an Arduino UNO development board. We implemented a specific visual task in VAM paradigm utilizing two arm lengths. This task is thereby referred as the Variable Spatial Length Maze (VSLM) in this project.

## VSLM training protocol and experimental paradigm

The mouse training process included three stages (Each stage has arm lengths set in random sequence): (1) Habituation phase (2–3 days) to introduce the maze, random arm lengths, and reward locations; (2) Training Phase I (8–10 days): Part 1 (2–3 days) introduced a visual cue at the end of the longer arm, guiding mice to use distance vision. In Part 2, cost of incorrect choices increased by closing the unselected arm after the first entry. (3) Training Phase II (2–4 days): The visual cue was removed, requiring mice to use depth cues.

## Behavioral tracking and quantification

Animal behavior trajectories were tracked using the Python-based deep convolutional neural network DeepLabCut (DLC) for image recognition [30]. Frames (100–150 frames) were extracted from 6–10 behavioral sessions and clustered by visual appearance using k-means clustering. The frames in the training set were manually labeled at key points on the animal's head, body center, and tail. A 50-layer pre-trained convolutional neural network (ResNet-50) was used and trained over 400,000 iterations. Trajectory accuracy was manually verified through labeled videos.

For VSLM behavioral data, the time points were standardized by marking the point when the mice crossed the junction between the start box and the maze arm as time point 0. A sliding time window of 5 frames was used to calculate the Euclidean distance between two neighboring coordinate points and determine average velocity.

## Calcium imaging setup

Single-photon calcium imaging was performed using the Miniscope V4 system, mountable on freely moving mice [31]. Behavioral tasks were recorded using a monochrome camera (standard industrial or a Basler camera, 50 fps) fixed above the maze. The open-source software Bonsai aligned timestamps to synchronize calcium signals and behavioral recordings [32].

## Calcium imaging data preprocessing

Calcium imaging raw video resolution was 608 × 608 pixels. $Ca^{2+}$ transient activity was extracted using Minian [33], an open-source Python package. Session-specific parameters were applied in the Minian Notebook due to the variable data quality. Spatial footprints and temporal signals of potential cells were iteratively identified using the constrained non-negative matrix factorization (CNMF) algorithm. Possible duplicate cells were manual merged, and noisy signals mistaken for cells were removed. Minian Cross-Registration pipeline was used to track and index cells across sessions or behaviors, which considers cells with footprints moving within 10 pixels as identical.

## Neuronal event rate calculation

Binarized events were extracted from the deconvolved activity trace of each neuron, marking timestamps of activity changes. Time points of these events were used to calculate the average cell firing rate with the tsgroup function in Pynapple [34]. A time bin of 1 second with 0.2-second step size was applied, and events within the time window was sliding-averaged to compute the firing rate for each cell. In VSLM analysis, the z-score of the firing rate was used to enable cell comparisons. Firing rates were normalized by trial count to accurately compare correct and incorrect trials, as the trial numbers varied.

## Area score calculation and depth preference analysis

Continuous data (e.g., cell traces) were binned along the target dimension (e.g., time, distance), and average cell activity was calculated per bin. Rate map for each cell was generated by calculating tuning curves of cell trace relative to position data at two depths, then dividing by occupancy data per bin. Depth preference differences were quantified by calculating the area score.

$$Area\ Score = (Ratemap1 - Ratemap2)/(|Ratemap1| + |Ratemap2|)$$

## Decoding analysis of neuronal activity

The decoding methods used in the research are modified by scikit-learn [35]. For each classifier, A random 80% of samples were used for training, and the remaining 20% for testing.

To decode the different depths in the visual cliff test, we employed the Gaussian Naive Bayes classifier. The decoding process was conducted independently for each group of neurons, with the classifier used to predict a binary outcome based on neuronal activity. Priors were adjusted based on data, assuming features likelihood followed a Gaussian distribution:

$$P(x_i|y) = \frac{1}{\sqrt{2\pi\sigma_y^2}} \exp\left(-\frac{(x_i - \mu_y)^2}{2\sigma_y^2}\right)$$

Parameters $\sigma_y$ and $\mu_y$ are estimated using maximum likelihood.

Decoding animal choices in the VSLM task was performed using the k-nearest neighbors (k-NN) algorithm implemented in scikit-learn. Five nearest neighbors were assigned to each query point, with uniform weights given to each

neighbor. The algorithm used (BallTree, KDTree, or Brute) was selected based on the training dataset, using Euclidean distance for computation.

The random forest classifier was used to decode animal position on the linear track and VSLM arms. The average decoding error was determined by iterating the random sampling process 10 times. The forest consisted of 100 trees, with nodes expanded until all leaves were pure or contained fewer than two split samples. A minimum of one sample per leaf node was required, with equal weight assigned to all samples.

## Mutual information analysis for neuronal activity

Cell trace was treated as continuous data, and head orientation was binarized into correct and incorrect arm. We used the function from scikit-learn based on nonparametric methods to estimate the mutual information for continuous target variables. This function employs a non-parametric method based on entropy estimation using k-nearest neighbors, with 3 neighbors selected for mutual information estimation.

Cells were binarized into discrete data. Mutual information for the discrete data was calculated and compared to shuffled data as a baseline. The mutual information was calculated as follows:

$$I(X, Y) = \sum_{x \in S_X} \sum_{y \in S_Y} p(x, y) \log \frac{p(x, y)}{p(x)p(y)}$$

x,y represent different sessions respectively.

Joint mutual information was derived from mutual information and conditional mutual information. The formula for conditional mutual information is as follows:

$$I(X, Y|Z) = \sum_{x \in S_X} \sum_{y \in S_Y} \sum_{z \in S_Z} p(x, y, z) \log \frac{p(x, y|z)}{p(x|z)p(y|z)}$$

x, y, z represent different sessions respectively.

The formula for calculating joint mutual information is as follows:

$$I(X, Y, Z) = I(X, Y) - I(X, Y|Z)$$

The stability of the depth-sensitive cells in the decision-making phase of VSLM test (DSVA cells) was assessed by comparing joint mutual information across three VSLM sessions with shuffled data.

## Data shuffling

To preserve data structure, shuffling did not involve full randomization. Instead, shuffled data in this study was generated by reordering the samples using random numbers, while maintaining certain aspects of internal data structure. This rearrangement process was repeated over several iterations to produce the final shuffled dataset.

## Statistical analysis

All experimental data were analyzed following random allocation, with statistical analyses and plots generated using GraphPad Prism 10.1.2. Unless specified, paired-sample t-tests compared mean differences in the same group, and independent-sample t-tests compared means between groups. For paired samples without normal distribution, the Mann-Whitney test was employed. Similarly, for two groups without normal distribution, the Wilcoxon matched-pairs signed rank test was applied. Pearson correlation analysis was used to assess the relationship between two variables

with normal distributions, and Spearman for non-normal distributions. A p-value of less than 0.05 considered statistically significant. The sample size, statistical parameters, and methods for each experiment are specified in the figure legends.

## Results

### Calcium imaging of V1 Activity in Depth-Related Behavioral Tasks

To simulate the real depth perception animals experience in nature, we utilized the Miniscope V4 to examine neuronal activity in freely moving animals. (Fig 1A-C) Behavioral training and calcium imaging recordings commenced two to three days later, alongside the modulated visual cliff test. The entire process took at least two months

We processed the raw videos (Fig 1D) using open-source pipeline Minian to extract pure cellular signals (Fig 1E). The sampled cells provided three main outputs—footprints, cell traces, and events (Fig 1F-G). Despite the temporal separation between behaviors, the number of extracted cells across different behaviors remained consistent (Fig 1H). Both the mean magnitude of cell activity and event rates were similar between the visual cliff and VSLM (Fig 1I-J), confirming the stability of the imaging preprocessing.

### Visual cliff test and depth-sensitive neuronal activity in V1

To track neuronal activity during depth perception, we used the visual cliff test, a classic depth-related paradigm. We further conducted two sessions with distinct depth comparisons to explore the effects of different depth stimuli on animals during the test. We compared the cumulative distance traveled on the deep side as a proportion of the total distance, at depths of 30 cm and 90 cm respectively (Fig 2A). The percentage of distance traveled by mice at 90 cm was significantly lower than at 30 cm depth, suggesting that they distinguish differences in depth across sessions. Additionally, for the deep side, the exploration of the center zone was significantly higher at 90 cm than at 30 cm (Fig 2B). A comparison of median speed at each frame before different time points also showed significant differences between depths (Fig 2C).

As the mice passively received depth information in the visual cliff, we aimed to identify neurons in the primary visual cortex that distinctly responded to depth changes. To calculate average firing rate at specific locations, tuning curves were normalized by position to control for its influence on neuronal firing. Neurons with area scores beyond the threshold of the mean ± standard deviation of all area score data were defined as Depth-sensitive cells in visual cliff test (DSVC cells) (Fig 2D). Ultimately, about one-third of cells from visual cliff recordings passed the criterion (Fig 2E). We compared the correlation between average firing rates at the two depths for both DSVC cells and rejected cells. A statistically significant difference in $R^2$ values of the fitted curves (Fig 2F) confirmed that greater depth sensitivity in DSVC cells compared to rejected cells.

Although the DSVC cells exhibited greater sensitivity than the rejected cells at both depths, we sought to ascertain whether there was a difference in the neural activity between these two cell types when encoding actual depth locations. First, we compared the performance of several classifiers (Fig 2G). Based on the calibration, the Gaussian Naive Bayes classifier, was closest to the ideal prediction curve and selected for best performance. Next, we calculated tuning curves of the DSVC cells and rejected cells along the X-axis of the maze (Fig 2H), which is perpendicular to cliffside. DSVC cells showed significantly higher decoding accuracy than rejected cells. Notably, the decoding accuracy difference was more pronounced on the shallow side of the maze.

### VSLM task and neuronal activity in active depth-based decision-making

The visual cliff test capture differences in neuronal firing patterns of depth perception. However, free-moving mice may not consistently focus on deep perception due to lack of task purpose. To motivate active depth cue use, we developed a depth-judgment paradigm to observe neuronal response. We designed the variable-arms maze (VAM) to create a spatial working memory task for mice (Fig 3A). Mice were trained to choose the longer arm for a food reward, then return to the

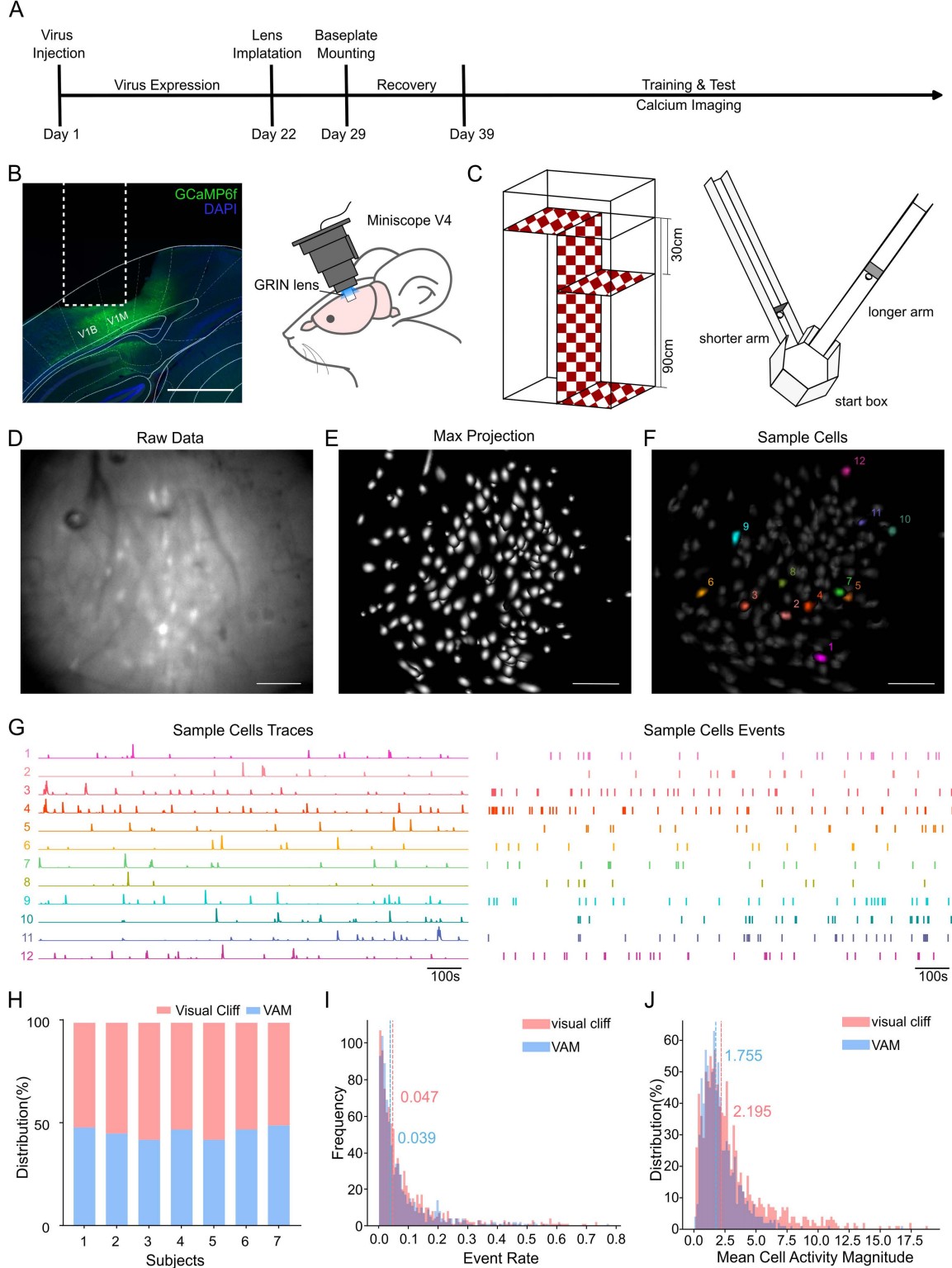

**Fig 1. In vivo calcium imaging recordings using the Miniscope in depth-related behaviors.** (A) Experimental timeline illustrating the calcium imaging surgery and data recording process. (B) GCaMP expression (green) and DAPI stained nuclei (blue) in V1 (reference line from: Paxinos and Franklin's The Mouse Brain In Stereotaxic Coordinates 5th edition) following recording procedure in an example animal. Scale bar, 1000 μm. The dashed

line indicates the position that the GRIN lens occupied (left). Schematic representation of in vivo calcium imaging(right). GCaMP6f is expressed in the primary visual cortex (V1), and the Miniscope is fixed on the mouse's head during recordings. (C) Schematic of the experimental setup of visual cliff (left) and VSLM (right). (D) An example frame of an imaging recording. Raw data were saved as a video. Scale bar, 100 μm. (E) Maximum projection of cell activity from an example session. (F) Twelve sampled cells were highlighted within the field of view, corresponding to the processed data shown in G. (G) Cell traces of the sampled cells, deconvolved from the original cell activities(left). Events detected from the cell traces of the sampled cells(right). (H) The number of detected cells in VSLM and the visual cliff tasks was comparable. (I) Distribution of event rates across all mice in the visual cliff and VSLM tasks. The dashed lines represent the mean event rates for each task, respectively. (J) Distribution of mean cell activity, calculated based on the magnitude of activity. The dashed lines indicate the mean values of overall mean cell activity.

start box. Seven mice completed the protocol, all showing improved accuracy during Training Phase II (Fig 3B). Behavioral recordings show that mice in the start box often turned to face different arms after doors opened (S1 File). These behaviors suggest a possible decision-making process before arm entry.

During decision-making, mouse behavior was categorized by head orientation: toward the correct or incorrect maze arm (Fig 3C). For each cell, we calculated the mutual information values between its calcium activity and the corresponding maze arm orientation at specific time points during both types of behaviors. Cells outside the 95% confidence interval of the shuffled data were defined as DSVA cells (Fig 3D-E), which were considered to be more specific to the depth-based discrimination task. To validate mutual information values for identifying cells, cells were ranked by their mutual information values. We then calculated the z-scores of event firing rates for the sorted DSVA cells and rejected cells before and after the time point 0, comparing correct and incorrect trials (Fig 3F).

To clarify firing patterns of these cells, we extracted fitted curves at the time points where firing rates exceeded 75% for DSVA cells and rejected cells. The fitted curves were transposed to interpret curve intercepts practically (Fig 3G). And the results are consistent with the conjecture that the DSVA cells encode information during depth-related tasks when mice make correct judgments. Also, rejected cells showed difference in firing rates between correct and incorrect trials. Additional analyses of fitted curve parameters across samples further supported using mutual information values to DSVA cells based on decision-period activity (S2 Fig). This validation also suggests that neurons in the primary visual cortex may play a role not only in receiving passive depth-related stimuli but also in active depth discrimination tasks. To determine whether DSVA cells were involved in active depth decision-making process, we trained k-nearest neighbors classifier to decode correct or incorrect selections based on the activity of DSVA cells and rejected cells. Decoding accuracy for DSVA cells and rejected cells was then compared over time (Fig 3H). The results showed that the decoding accuracy for DSVA cells significantly increased before time point 0 and rose again after the mice entered the maze arm. The peak decoding accuracy was significantly higher for DSVA cells than for rejected cells (Fig 3I). Thus, in the depth-based discrimination task, neurons in the primary visual cortex appear involved in active depth-based decision-making processes.

## Properties and stability of V1 neuronal populations in depth-related tasks

Since VSLM behavior can be repeatedly measured, we tested the stability of DSVA cells across days by cross-registration, and calculated the DSVA cells overlapped across three VSLM sessions. The joint mutual information for the cells across three sessions was significantly higher than shuffled data, indicating stable and reliably identifiable DSVA cells (S3 Fig).

To test if DSVC cells and DSVA cells from a uniform population, we used the same method to examine overlap (Fig 4A-C). Actual mutual information showed no significant difference from the shuffled data (Fig 4C). Thus, cells involved in visual cliff test and VSLM appears to be distinct.

As mice move forward in the maze arm, depth information their received continuously changes. However, it remains unclear which specific neuronal populations are engaged after the mice enter the arm (Fig 4D). Since primary visual cortex neurons respond to immediate visual stimuli, it is important to see if they respond to changing depth. To investigate this, we trained random forest classifier on neuronal activity tuning curves related to mouse positions in the VSLM maze

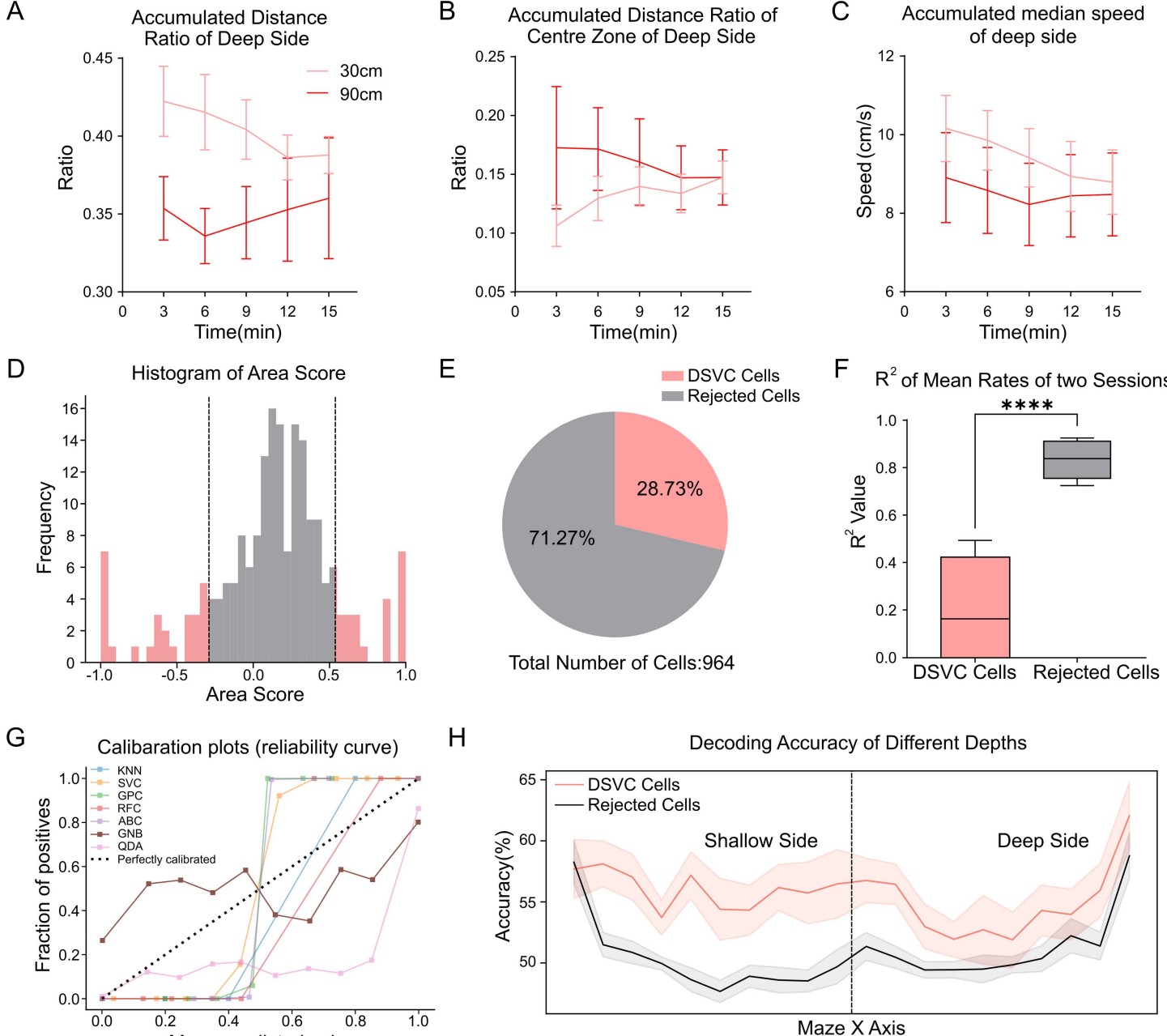

**Fig 2. Identification of cells specific to visual cliff test.** (A) Accumulated distance ratio on the deep side of all zones in the 30 cm visual cliff session is significantly larger than in the 90 cm session across time periods. Error bars represent SEM. Two-way ANOVA, ****p < 0.0001. (B) The accumulated distance ratio in the center zone of the deep side during the 30 cm visual cliff session is significantly smaller than in the 90 cm session across time periods. Error bars represent SEM. Two-way ANOVA, ****p < 0.0001. (C) The accumulated median speed on the deep side during the 30 cm visual cliff session is significantly higher than in the 90 cm session across different time periods. Error bars represent SEM. Two-way ANOVA, ****p < 0.0001. (D) Histogram of area scores for all neurons from a sample subject. The black dashed lines indicate the thresholds for the area scores. The sample DSVC cells are represented by red, and the rejected cells are represented by gray. (E) Distribution of DSVC cells and rejected cells for all subjects (n = 7): DSVC cells make up 28.73% of recorded neurons, and rejected cells 71.27% (total 964 neurons). (F) Comparison of R² values for fitted curves of mean rates in different sessions between DSVC cells and rejected cells. R² value for DSVC cells is significantly lower than for rejected cells. Paired t-test, ****p < 0.0001. (G) Calibration plot for different classifiers used to decode the two depths, with each color representing a different classifier (KNN, k-nearest neighbors; SVC, C-Support Vector Classification; GPC, Gaussian process classifier; MLP, Multi-layer Perceptron classifier; RFC, random forest classifier; ABC, AdaBoost classifier; GNB, Gaussian Naive Bayes; QDA, QuadraticDiscriminantAnalysis), reduced transparency for classifiers not selected, and lines showing

classifier reliability (actual depth fraction vs predicted). The black dashed line represents a perfectly reliable curve. (H) Decoding accuracy for the two depths (30 cm and 90 cm) between DSVC cells and rejected cells along the X-axis of the maze (perpendicular to the cliffsides). The GNB classifier was used, with the black dashed line indicating cliffsides and solid lines showing mean decoding accuracy along the x-axis for DSVC cells and rejected cells. Shading represents mean ± SEM. DSVC cells show significantly higher decoding accuracy than rejected cells. Two-way ANOVA, ****$p < 0.0001$.

(Fig 4E). Compared to shuffled data, DSVA cells showed significantly lower decoding errors, and higher prediction probabilities (S3 Fig). The cumulative fraction of decoding error showed a median of 1.25 cm for DSVA cells. Thus, the activity of DSVA cells reliably predicts the animal's actual spatial location.

Since DSVA cells successfully predicted, we questioned if DSVC cells could do the same. To explore this, we compared DSVC cells and DSVA cells after cross-registration. Interestingly, DSVC cells showed significantly smaller decoding errors than DSVA cells in position prediction within the VSLM maze (Fig 4F), though their prediction probabilities did not differ significantly (Fig 4G). DSVA cells had higher decoding errors than those original cells, likely due to reduced cell numbers post-cross-registration. Median decoding error was lower for DSVC cells than DSVA cells (Fig 4H). Thus, DSVC cells appear to predict the linear position of the mice even more accurately than DSVA cells.

### Linear track test and positional encoding in V1

Though the VSLM maze arm can be seen as a linear track with varying visual stimuli, cognition is affected by factors like differing visual information between round trips, track length limits, and variable arm lengths. To clarify how primary visual cortex neurons relate to spatial depth parameters, we characterized cell activity in a linear track.

In this task, animals ran back and forth in the linear track to obtain food at both ends. The animal's coordinates were converted to relative positions, assuming constant forward movement. We then used a random forest classifier to decode neural activity from linear track, cross-registered beforehand with DSVA cells and DSVC cells. The classifiers were trained using tuning curves of calcium activity relative to both actual and relative mice positions, in the linear track respectively. (Fig 5A). Decoding errors for actual positions were significantly smaller than those for transformed (relative) positions across cell groups (Fig 5B).

This shows that decoding errors for actual positions were more evenly distributed between positive and negative values. In contrast, decoding errors for relative positions were predominantly positive, suggesting decoded positions were closer to the target ahead than to the mice's actual location. By analyzing the ratio of positive to negative values for each data group, we found that positive values were more frequent when decoding for relative positions (Fig 5E), suggesting that neurons encode relative positions nearer to the target, or expected positions, rather than the mouse itself.

Due to the high decoding accuracies, the median decoding errors were frequently zero. We compared average decoding errors across all subjects for each data group (Fig 5D). Results showed significantly higher decoding accuracy for actual positions than for relative positions, with no significant difference in decoding error between DSVC cells and DSVA cells. Additionally, classifier decoding probabilities (Fig 5C) showed a significant difference between actual and relative positions for DSVA cells, but not for DSVC cells.

### Discussion

This study reveals the primary visual cortex (V1) as a dynamic processor of depth information, capable of adapting its role based on behavioral context. We investigated V1 neuronal activity in both passive and active depth-perception scenarios, discovering distinct neuronal populations engaged in different aspects of depth processing. Specifically, our results demonstrate that while one subset of V1 neurons encodes depth information in response to passive, non-directed stimuli, another group is selectively active during decision-making tasks that require active depth discrimination. These findings underscore a functional segregation within V1 that is indicative of its complex role beyond basic visual processing.

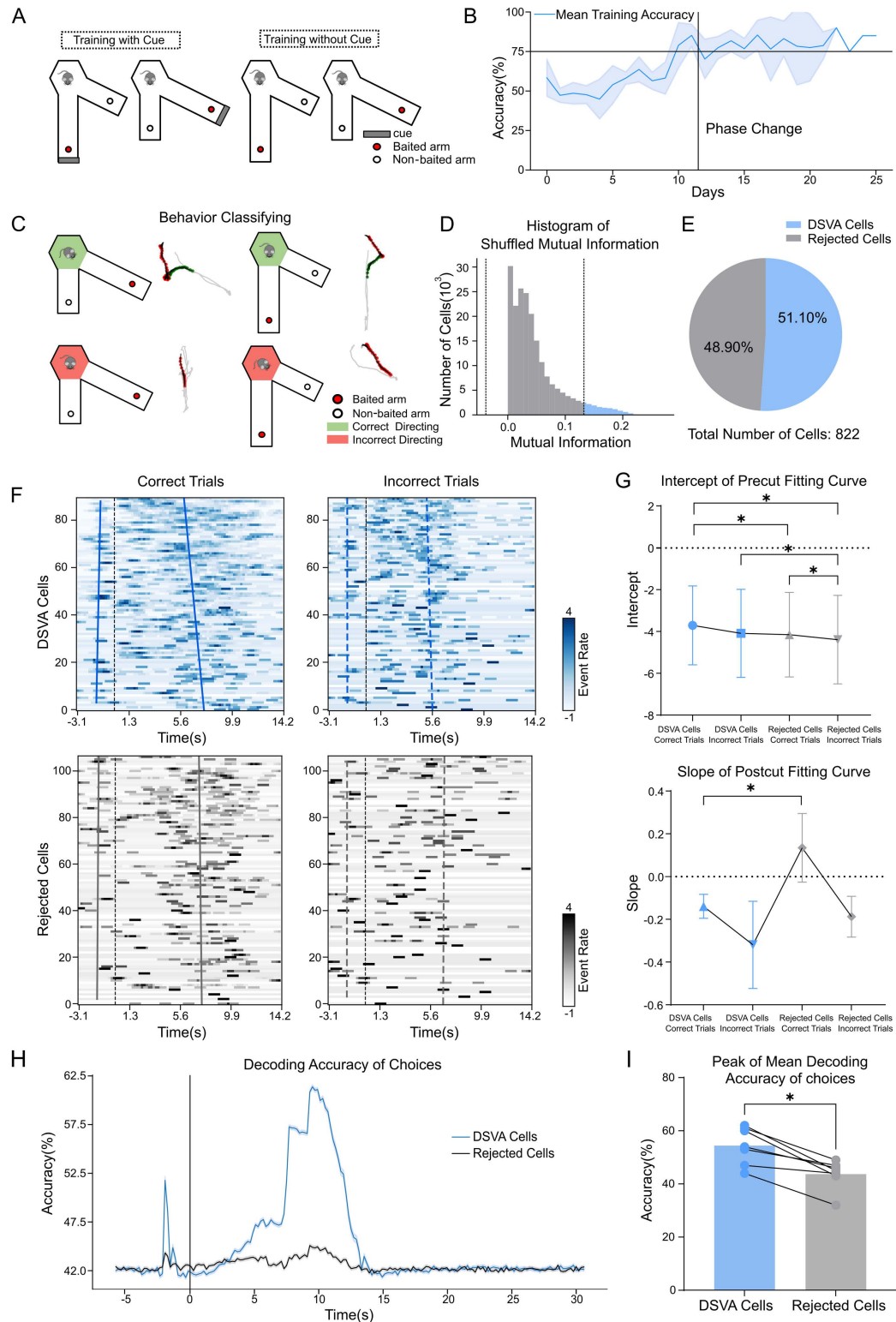

**Fig 3. Identification of cells specific to active depth discrimination.** (A) Schematic of the VSLM training paradigm. In the first training phase (left), a cue card and food reward are placed at the end of the correct, longer arm. In the second phase (right), no cue card is used, but the food reward remains in the correct (longer) arm. (B) Training accuracy over time, divided into two phases. The black dashed line marks the phase transition; blue line shows

individual subject accuracy, and the blue shade shows standard deviation of the accuracy across subjects (n=7). (C) Schematic of behavioral classification in the VSLM task, categorizing mouse head orientation in the start box toward either the correct or incorrect arm. (D) Histogram of shuffled mutual information values for all cells in a sample subject. The black dashed lines show the 95% confidence interval; DSVA cells are represented by blue, rejected cell are represented by gray. (E) Distribution of DSVA cells (51.10%) and rejected cells (48.90%) across all subjects in the VSLM task. A total of 822 neurons were recorded, averaging across three sessions per subject. (F) Heatmap of event rates of ordered cells, ranked by mutual information value (DSVA cells in blue, rejected cells in gray) for correct (left) and incorrect (right) trials in a sample subject. Each row represents a single cell with color intensity indicating event rate z-score on the color bar. The black dashed lines indicate time point 0 (mouse entering the arm), blue lines show fitted curves for event rates of DSVA cells above 75% before and after time point 0 in correct trials, and gray lines show rejected cells in correct trials, blue dashed lines show DSVA cells in incorrect trials, gray dashed lines show rejected cells in incorrect trials. (G) Intercepts of the fitted lines before time point 0 and slopes after time point 0 across all subjects, with markers representing categories. Error bars show SEM for all subjects. Four combined categories show significant differences in intercepts before time point 0, with DSVA cells in correct trials being significantly different from rejected cells in correct trials. Mann-Whitney test, *p<0.05. (H) Decoding accuracy of choices across all trials in a sample session for DSVA cells and rejected cells. The lines represent decoding accuracy over time; shaded areas indicating mean±SEM. (I) Peak mean decoding accuracy for choices in the VSLM task across all subjects, comparing DSVA cells and rejected cells. Each dot represents a subject; DSVA cells have significantly higher peak accuracy than rejected cells Wilcoxon test, *p<0.05.

## Passive depth perception and context-dependent neural encoding

Depth information is crucial for animals to perceive and navigate the scale of three-dimensional space [1], rising questions about how it is received and processed. In the visual cliff paradigm, a well-established model for studying depth perception, we identified neurons in V1 that passively respond to depth cues, even in the absence of a targeted behavioral objective. This response mirrors real-world settings where animals passively gather spatial information as they navigate their environment [36]. This observation aligns with findings in other sensory modalities, where cortical areas respond selectively to environmental context cues, enabling efficient sensory integration [37,38].

## Active depth discrimination and cognitive engagement in V1

While depth is a key type of spatial information [39], how do neuronal patterns change when depth becomes the main criterion for decision-making? Few behavioral paradigms are available to study depth cognition, with varying effectiveness in probing depth-related neuronal patterns. For example, popular paradigms like the visual water task [40] and jumping experiments [41,42] restrict certain in vivo recording techniques. In this study, we designed the VSLM task, to examine active depth perception, required animals to utilize depth information actively in decision-making, rewarding correct distance judgments. Our analysis showed that a distinct group of V1 neurons was preferentially active during the decision-making phase of this task, revealing a form of active, depth-dependent coding. These depth-sensitive cells exhibited specific patterns of neural activity corresponding to correct versus incorrect choices, supporting the notion that V1 contributes to cognitive processes that extend beyond simple sensory reception. This active engagement of V1 in a task-driven context indicates that, in addition to providing a visual spatial map, V1 may play a role in higher-order processes, such as decision-making and spatial memory formation.

## Functional segregation and cross-task consistency in depth-related processing

By comparing depth-sensitive neuronal populations across both tasks, we observed a notable lack of overlap between the passive and active depth-responsive cells, suggesting a form of functional segregation within V1. These findings point to specialized V1 subpopulations that may be independently recruited based on task demands, possibly facilitating efficient, context-specific processing. Such functional segregation within V1 might reflect its broader role in multisensory integration and spatial navigation, supporting the idea that primary visual cortex processing is not merely static but highly adaptive [43,44].

Additionally, the neurons from different groups continue encoding the animal's location on the linear track even after decision-making is completed. These findings imply that V1 neurons are capable of "dual-mode" functionality—one mode of passive stimulus reception and another of active decision-making. This parallels findings in other brain regions, which

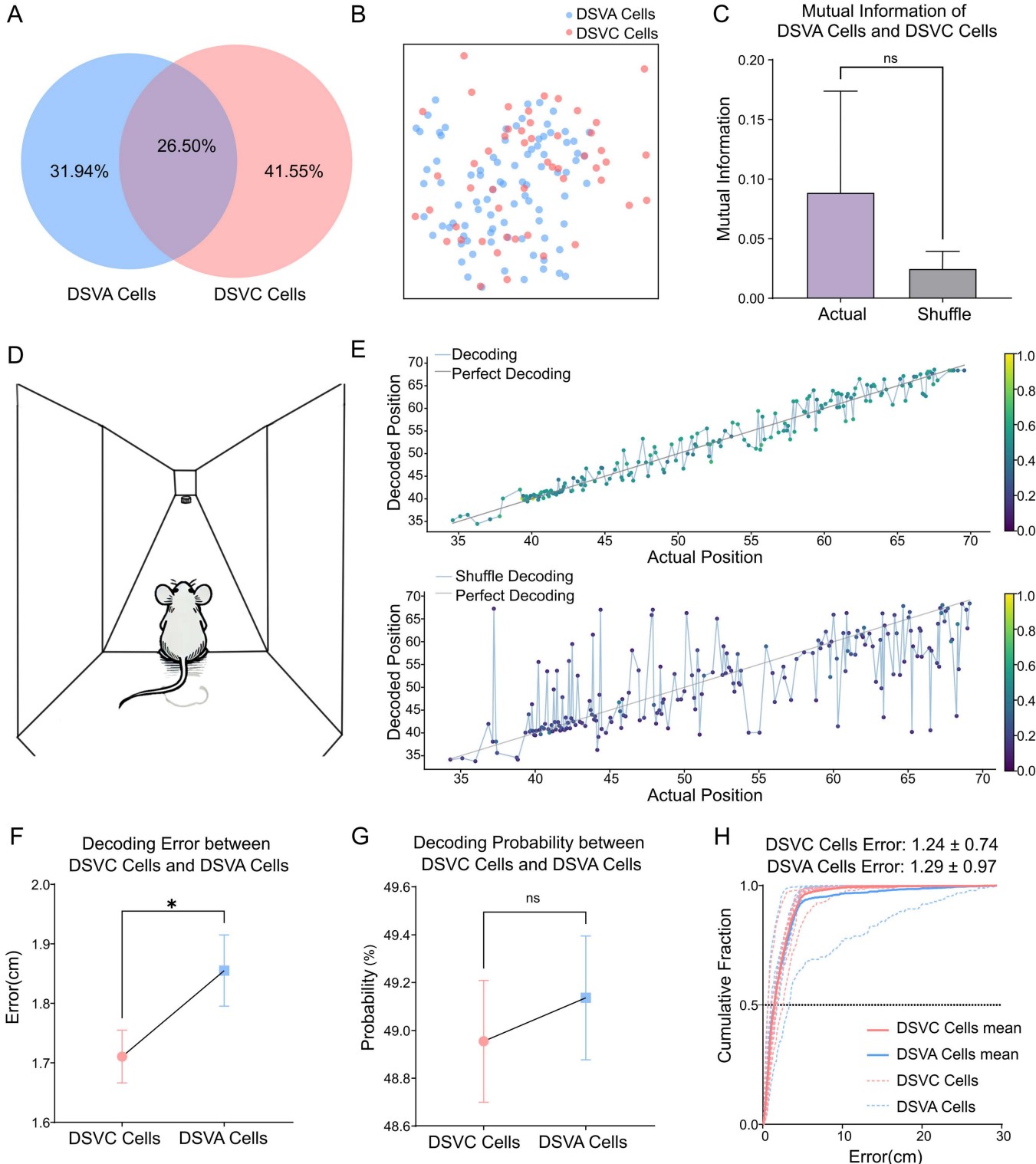

**Fig 4. Prediction of animal position in VSLM arms using different cell groups.** (A) Venn diagram illustrating the overlap of crossed cells between VSLM and visual cliff tasks, with 375 crossed cells. (B) Map of DSVC cells and DSVA cells for a sample subject; each dot represents a cell. (C) Mutual information between DSVA cells and DSVC cells compared to shuffled data. Error bars represent the STD of individual data. Mutual information of actual data shows no significant difference from the shuffled data. Wilcoxon test, not significant (ns), p>0.05. (D) Schematic of decoding stage in the VSLM

task for mouse subjects. (E) Predicted versus actual positions decoded from filtered cell activities and shuffled data in a sample subject. Each dot represents a decoded result at a test position, with colors indicating decoding probability on the color bar. The black dashed line represents a perfect decoding result. (F) Prediction errors for positions decoded by DSVC cells and DSVA cells; DSVC cells errors are significantly smaller. Mann-Whitney test, *p < 0.05. (G) Decoding probabilities of predicted positions between DSVC cells and DSVA cells show no significant. Mann-Whitney test, not significant (ns), p > 0.05. (H) Cumulative fraction of decoding errors across subjects (n = 7) for DSVC cells and DSVA cells. Median decoding error is 1.24 ± 0.74 cm for DSVC cells and 1.29 ± 0.97 cm for DSVA cells. Dashed line show median cumulative fraction of decoding errors.

exhibit similarly adaptive responses based on behavioral relevance [45,46]. As a result, this adaptive functionality within V1 could enhance an animal's ability to interpret and act upon depth-related information in complex environments, such as navigating obstacles or assessing threats.

However, the passive-depth-sensitive neurons encoded locations more accurately than active depth-discrimination neurons, indicating functional overlap and dissociation within the primary visual cortex. Even when engaged in a same function, neurons may encode other processes in parallel, leading to population differences in firing patterns. The distinction between passive depth encoding and active depth-based decision-making in V1 may suggest a possible information flow loop among primary visual cortex neurons after active decision-making or an expanded role for the primary visual cortex beyond visual processing, considering multiple layers of information flow and microcircuits in visual cortex [47–49].. Also, this functional adaptability suggests that V1 may transition between encoding modes depending on the demands of the environment and task at hand [50–52], potentially forming a basis for how animals navigate complex, multi-dimensional spaces.

### Implications for visual cortex functionality and prospective coding

In our study, both decision-related and passive depth-sensitive neurons encoded the animal's actual position on the linear track, as opposed to the subjective position, supporting location-specific cells in the primary visual cortex. The ability of V1 to encode allocentric (environment-centered), rather than egocentric (body-centered) information, suggests that it may contribute to spatial processing networks involving the hippocampus, which is known to encode space and navigate based on both perspective and relational cues [13,14,16]. This encoding capacity may suggest that the depth information has evolved to more delicate spatial information in V1, allowing V1 to act as a "pre-filter" for relevant spatial information before it is relayed to higher-order cortical areas for further cognitive processing.

Finally, our findings on V1's potential for prospective coding—encoding locations relative to an anticipated position rather than only present spatial coordinates—provides new insights into how V1 might support forward-looking functions. Neurons involved in the active depth discrimination task showed a tendency to encode prospective spatial information, aligning with previous reports of prospective and retrospective coding in spatial memory networks like the hippocampus [53–55]. This suggests that V1 may serve a predictive function, assisting animals in preparing for upcoming spatial decisions by pre-encoding anticipated depth cues.

These prospective characteristics add a valuable layer to our understanding of V1's function, positioning it as an active participant in complex, forward-planning behaviors essential for effective navigation. This finding, alongside the identified functional segregation, underscores V1's role as a context-responsive, cognitive interface that provides more than merely sensory inputs.

### Conclusion and future directions

In sum, this study provides compelling evidence that the primary visual cortex serves multifaceted roles in depth perception. Our results support a model in which V1 can flexibly switch between passive sensory encoding and active cognitive engagement, adapting its functions to suit behavioral demands. Future work could further explore the network interactions between V1 and other spatially-oriented cortical areas, elucidating how V1 contributes to larger, cross-modal networks that underpin spatial memory and decision-making. Additionally, examining how these V1 neural populations adapt over

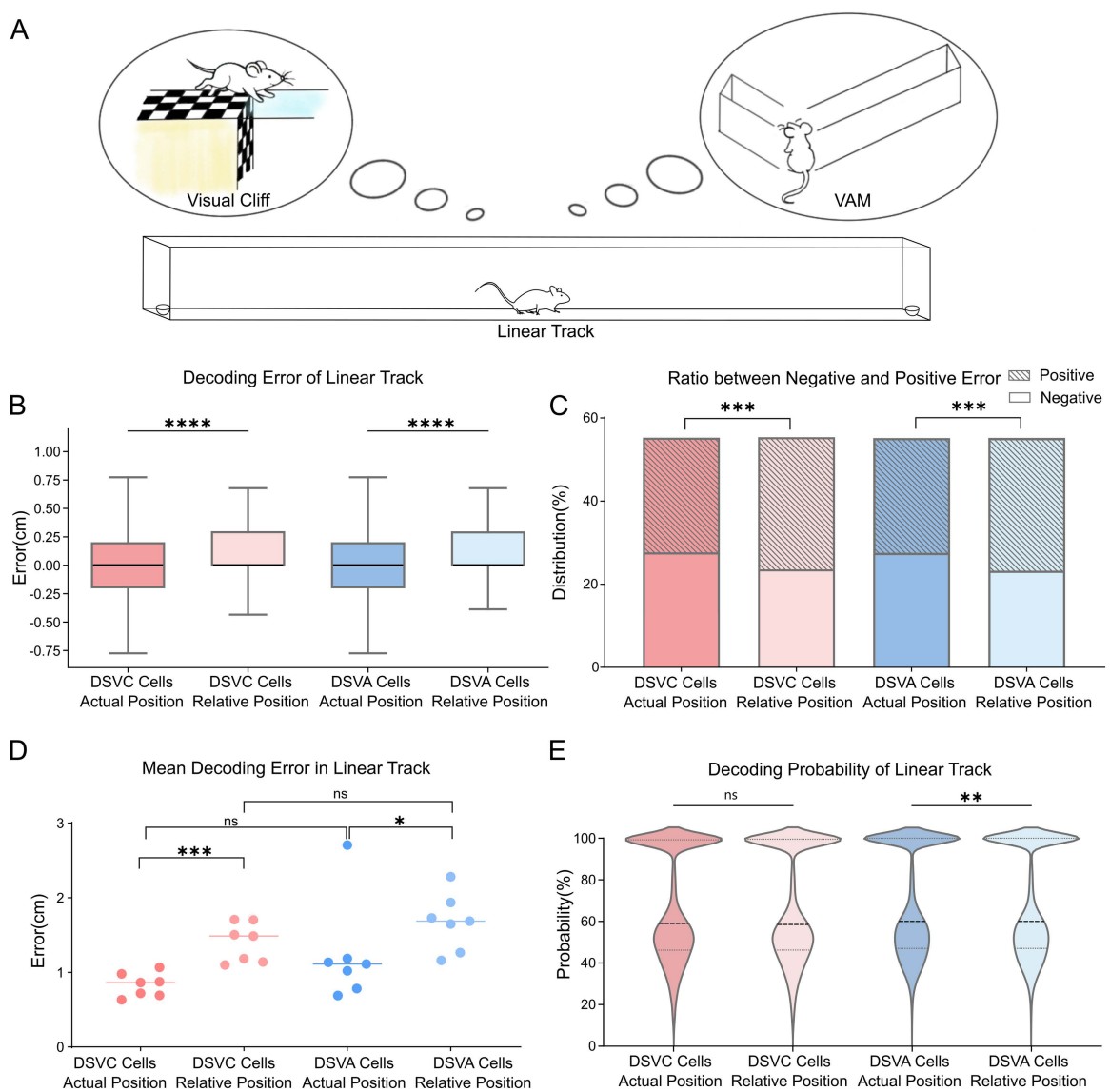

**Fig 5. Decoding performance of animal position on the linear track.** (A) Schematic of decoding process for animal position on the linear track using DSVA cells and DSVC cells. (B) Decoding errors for actual and relative positions on the linear track using DSVA cells and DSVC cells. Black lines indicated medians, all at 0. Decoding errors significantly differed between actual and relative positions for both DSVA cells and DSVC cells. Mann-Whitney test, ****$p < 0.0001$. (C) Ratio of negative to positive errors in all decoding errors on the linear track using DSVA cells and DSVC cells. Each bar represents both negative error ratio (blank) and positive error ratio (slashes). Statistical test was performed on the positive-to-negative ratio. Positive error ratio was significantly higher than negative error ratio for relative positions, with no significant difference between DSVA cells and DSVC cells. Mann-Whitney test, ***$p < 0.001$. (D) Mean absolute decoding errors for actual and relative positions on the linear track using DSVA cells and DSVC cells across subjects ($n = 7$). Absolute decoding errors for actual positions were significantly smaller than for relative positions for both cell types. No significant difference in mean absolute decoding errors was found between DSVA cells and DSVC cells. Mann-Whitney test, ***$p < 0.001$; *$p < 0.05$; ns, $p > 0.05$. (E) Decoding probabilities for actual and relative positions on the linear track using DSVA cells and DSVC cells, shown as violin plot. Black dashed lines indicate the medians. No significant difference was found for DSVC cells, but a significant difference was seen for DSVA cells. Mann-Whitney test, **$p < 0.01$; ns, $p > 0.05$.

learning or under varying depth-related challenges would shed light on their plasticity and long-term functional roles in spatial cognition.

## Supporting information

**S1 File. A sample correct trial of VSLM.**
(MP4)

**S2 Fig. Parameters of fitted curves of time points of event rate of cells sorted by mutual information. Four combined categories show significant differences by statistic markers. Different markers represent different categories. (A)** Slopes of the fitted lines before time point 0 for all subjects. Different markers represent different categories. Error bars represent SEM for all subjects. **(B)** Intercepts of the fitted lines before time point 0 for all subjects. Error bars represent SEM for all subjects. Mann-Whitney test, $*p < 0.05$. **(C)** R2 of the fitted lines before time point 0 for all subjects. Error bars represent SEM for all subjects. The top and bottom edges of the box represent the maximum and minimum value, respectively. Mann-Whitney test, $*p < 0.05$. **(D)** Slopes of the fitted lines after time point 0 for all subjects. Different markers represent different categories. Error bars represent SEM for all subjects. Mann-Whitney test, $*p < 0.05$. **(E)** Intercepts of the fitted lines after time point 0 for all subjects. Error bars represent SEM for all subjects. Mann-Whitney test, $*p < 0.05$. **(F)** R2 of the fitted lines after time point 0 for all subjects. Error bars represent SEM for all subjects. The top and bottom edges of the box represent the maximum and minimum value, respectively. Mann-Whitney test, $*p < 0.05$.
(TIF)

**S3 Fig. Cross-registration of cells in VSLM and prediction of animal position in VSLM arms. (A)** Cell maps showing unshifted and shifted cell positions for a sample subject. The unshifted map shows original cell positions across three VSLM sessions (left), while the shifted map shows positions adjusted by parameters calculated through cross-registration. Different colors indicate sessions. **(B)** Venn diagram illustrating the overlap of crossed cells across three VSLM sessions with 367 cells identified as crossed. **(C)** Positions of DSVA cells within crossed cells from three VSLM sessions for a sample subject.; each dot represents a cell. **(D)** Schematic of conditional mutual information for cells filtered across three VSLM sessions. $H(X \mid Y, Z)$ is conditional entropy of variable X given Y and Z; $I(X; Y \mid Z)$ is conditional mutual information between X and Y given Z. $I(X; Y; Z)$ is the joint mutual information of variables, X, Y, and Z. **(E)** Conditional mutual information of DSVA cells across three VSLM sessions compared to shuffled data. Error bars represent STD of individual data. The conditional mutual information of the actual data is significantly higher than shuffled data. Wilcoxon test, $*p < 0.05$. **(F)** Decoding errors of predicted positions for actual data vs. shuffled data; actual data are significantly smaller than shuffled data. Mann-Whitney test, $****p < 0.0001$. **(G)** Decoding probabilities for predicted positions in actual data vs. shuffled data; actual data probabilities are significantly higher. Mann-Whitney test, $****p < 0.0001$. **(H)** Cumulative fraction of decoding errors across subjects, with a median decoding error of $1.25 \pm 0.76$ cm. Dashed line show individual subjects; solid line represents mean cumulative fraction of decoding errors.
(TIF)

## Acknowledgments

The authors thank lab members Shuxin Li, Yongyi Luo and Yu Zhou for the testing and data collection.

## Author contributions

**Conceptualization:** Nuo Dong, Haibing Xu.

**Data curation:** Nuo Dong.

**Formal analysis:** Nuo Dong, Yuping Tan, Yuyuan Wang, Haibing Xu.

**Funding acquisition:** Haibing Xu.

**Investigation:** Nuo Dong.

**Methodology:** Nuo Dong, Yuping Tan, Yumin Chen.

**Project administration:** Haibing Xu.

**Resources:** Haibing Xu.

**Supervision:** Haibing Xu.

**Validation:** Haibing Xu.

**Visualization:** Nuo Dong.

**Writing – original draft:** Nuo Dong, Haibing Xu.

**Writing – review & editing:** Nuo Dong, Yuping Tan, Yuyuan Wang, Yumin Chen, Haibing Xu.

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
