## [Decision Letter · Decision Letter 0]

14 Apr 2025

PONE-D-24-51051Characterization of Depth Perception Information Inferred from Neuronal Activity in Primary Visual CortexPLOS ONE

Dear Dr. Xu,

Thank you for submitting your manuscript to PLOS ONE. After careful consideration, we feel that it has merit but does not fully meet PLOS ONE’s publication criteria as it currently stands. Therefore, we invite you to submit a revised version of the manuscript that addresses the points raised during the review process**Specifically, in the light of reviewers' comments and feedback appended below** .

We look forward to receiving your revised manuscript.

Kind regards,

Umer Asgher, PhD

Academic Editor

PLOS ONE

Journal Requirements:

 “This research was supported by the National Natural Science Foundation of China (Grant No. 32171038 and W243058 to H.X.), STI2030-Major Projects (Grant No. 2022ZD0205000 to H.X.), and Guangdong Science and Technology Department (2022A515011896 and 2024A1515011033 to H.X.).”        

Reviewers' comments:

Reviewer's Responses to Questions

**Comments to the Author**

1. Is the manuscript technically sound, and do the data support the conclusions?

Reviewer #1: Yes

Reviewer #2: Yes

Reviewer #3: Yes

2. Has the statistical analysis been performed appropriately and rigorously? 

Reviewer #1: Yes

Reviewer #2: Yes

Reviewer #3: Yes

3. Have the authors made all data underlying the findings in their manuscript fully available?

Reviewer #1: Yes

Reviewer #2: Yes

Reviewer #3: Yes

4. Is the manuscript presented in an intelligible fashion and written in standard English?

Reviewer #1: Yes

Reviewer #2: Yes

Reviewer #3: Yes

5. Review Comments to the Author

Reviewer #1: This paper reports investigations on how primate-V1 could play complex, higher-level role beyond our imagination upon its basic visual processing, i.e., passive and active depth-perception to constitute a functional segregation. The paper is well written and gradually illustrating the main results through experiments on mouse. The results are of importance to further understand the unexplored roles of V1 visual cortex in visual processing tasks. From my perspective, it is a high-quality paper deserves to be published in the journal of PLOS ONE.

Reviewer #2: Comments to the Author

In the current manuscript, Dong et al. investigated an important question: what are the specific contributions of neurons in primary visual cortex in various aspects of depth perception. In this manuscript, the authors trained mice to perform two tasks: visual cliff (passive depth perception) and variable-length-arm maze (active depth discrimination). They recorded in vivo calcium imaging in freely moving mice, and the constrained non-negative matrix factorization algorithm to identified depth-related V1 neurons in two tasks (DSVA and DSVC cells). They showed that specific groups of V1 neurons are selectively activated, suggesting functional segregation. Moreover, V1 neurons prefer encoding objective positions rather than egocentric distances in non-depth-based tasks.

The manuscript is written clear. The analyses are sound, and the results are generally interesting. I have however a few concerns, and hope the author will be able to clarify those issues.

Concerns:

1. The definition of DSVA cell was introduced at Page 11 Line 210, but DSVA cell firstly appeared at Page 18 Line 334. This may confuse the readers, please modify.

2. In Variable-Arms Maze (VAM) task, the number of arms can be changed. In this manuscript, the authors designed a radial variable-length-arm maze, required mice to utilize depth information actively in decision-making. This task was more specific, and should be distinguished from other VAM tasks. I suggest using VSLM, instead of VSM, to define this task.

Reviewer #3: Since I have suggested to accept the manuscript, I have no more review comments to the author.

I have no more explanation for my answers to the questions above or additional comments for the authors.

6. PLOS authors have the option to publish the peer review history of their article (what does this mean? ). If published, this will include your full peer review and any attached files.

**Do you want your identity to be public for this peer review?** For information about this choice, including consent withdrawal, please see our Privacy Policy .

Reviewer #1: No

Reviewer #2: No

Reviewer #3: No

---

## [Author Response · Author response to Decision Letter 1]

22 May 2025

Response to Reviewers' Comments

PONE-D-24-51051

Characterization of Depth Perception Information Inferred from Neuronal Activity in Primary Visual Cortex

Dear Reviewers,

Thank you for your valuable advice. Your suggestions have been extremely beneficial to us. We sincerely appreciate the time and effort you have dedicated to reviewing our manuscript. Below, we provide a point-by-point response to each concern raised.

Reviewer #1

This paper reports investigations on how primate-V1 could play complex, higher-level role beyond our imagination upon its basic visual processing, i.e., passive and active depth-perception to constitute a functional segregation. The paper is well written and gradually illustrating the main results through experiments on mouse. The results are of importance to further understand the unexplored roles of V1 visual cortex in visual processing tasks. From my perspective, it is a high-quality paper deserves to be published in the journal of PLOS ONE.

We are deeply grateful for the reviewer’s positive evaluation and endorsement of our study. The acknowledgment of the paper’s quality and relevance to understanding V1’s higher-level roles is highly encouraging.

Reviewer #2

In the current manuscript, Dong et al. investigated an important question: what are the specific contributions of neurons in primary visual cortex in various aspects of depth perception. In this manuscript, the authors trained mice to perform two tasks: visual cliff (passive depth perception) and variable-length-arm maze (active depth discrimination). They recorded in vivo calcium imaging in freely moving mice, and the constrained non-negative matrix factorization algorithm to identified depth-related V1 neurons in two tasks (DSVA and DSVC cells). They showed that specific groups of V1 neurons are selectively activated, suggesting functional segregation. Moreover, V1 neurons prefer encoding objective positions rather than egocentric distances in non-depth-based tasks.

The manuscript is written clear. The analyses are sound, and the results are generally interesting. I have however a few concerns, and hope the author will be able to clarify those issues.

We thank the reviewer for their thorough and insightful comments. Below are our detailed responses:

Comment 1: The definition of DSVA cell was introduced at Page 11 Line 210, but DSVA cell firstly appeared at Page 18 Line 334. This may confuse the readers, please modify.

Response: We agree that the initial presentation of DSVA cells could cause confusion. In the revised manuscript, we introduced the term "DSVA cells" with its definition when first mentioned (Page 12, Line 217).

Comment 2: In Variable-Arms Maze (VAM) task, the number of arms can be changed. In this manuscript, the authors designed a radial variable-length-arm maze, required mice to utilize depth information actively in decision-making. This task was more specific, and should be distinguished from other VAM tasks. I suggest using VSLM, instead of VSM, to define this task.

Response: We appreciate this suggestion. To avoid ambiguity with general VAM tasks, we replaced all statements of "VAM" with "VSLM" (Variable Spatial Length Maze) throughout the text.

And we added an explanation in the Methods section (Page 7, Line 125) distinguishing VSLM from other VAM paradigms.

Reviewer #3

Since I have suggested to accept the manuscript, I have no more review comments to the author.

I have no more explanation for my answers to the questions above or additional comments for the authors.

We sincerely thank the reviewer for the supportive assessment and recommendation for acceptance.

We hope these revisions and responses have adequately addressed all concerns. Thank you again for the opportunity to improve our work. We look forward to your further guidance.

Sincerely,

Dr. Haibing Xu

Professor

Guangdong-Hong Kong-Macao Greater Bay Area Center for Brain Science and Brain-Inspired Intelligence

Southern Medical University, Guangzhou, China

Email: haibingxu@smu.edu.cn

Response to Editor's Comments

PONE-D-24-51051

Characterization of Depth Perception Information Inferred from Neuronal Activity in Primary Visual Cortex

Dear Editor,

We sincerely appreciate the time and effort you have dedicated in our manuscript.

We have revised the format of this manuscript and the statements of the financial disclosure, and provided the data availability. In addition, the figures of the study have been corrected by PACE.

We hope these revisions and responses have adequately addressed all concerns. Thank you again for the opportunity to improve our work. We look forward to your further guidance.

Sincerely,

Dr. Haibing Xu

Professor

Guangdong-Hong Kong-Macao Greater Bay Area Center for Brain Science and Brain-Inspired Intelligence

Southern Medical University, Guangzhou, China

Email: haibingxu@smu.edu.cn

---

## [Decision Letter · Decision Letter 1]

22 Jul 2025

Characterization of Depth Perception Information Inferred from Neuronal Activity in Primary Visual Cortex

PONE-D-24-51051R1

Dear Dr. Xu,

We’re pleased to inform you that your manuscript has been judged scientifically suitable for publication and will be formally accepted for publication once it meets all outstanding technical requirements.

Kind regards,

Rongchun Han

Academic Editor

PLOS ONE

Reviewers' comments:

Reviewer's Responses to Questions

**Comments to the Author**

1. If the authors have adequately addressed your comments raised in a previous round of review and you feel that this manuscript is now acceptable for publication, you may indicate that here to bypass the “Comments to the Author” section, enter your conflict of interest statement in the “Confidential to Editor” section, and submit your "Accept" recommendation.

Reviewer #1: All comments have been addressed

Reviewer #2: (No Response)

2. Is the manuscript technically sound, and do the data support the conclusions?

Reviewer #1: Yes

Reviewer #2: (No Response)

3. Has the statistical analysis been performed appropriately and rigorously? 

Reviewer #1: Yes

Reviewer #2: (No Response)

4. Have the authors made all data underlying the findings in their manuscript fully available?

Reviewer #1: Yes

Reviewer #2: (No Response)

5. Is the manuscript presented in an intelligible fashion and written in standard English?

Reviewer #1: Yes

Reviewer #2: (No Response)

6. Review Comments to the Author

Reviewer #1: (No Response)

Reviewer #2: (No Response)

7. PLOS authors have the option to publish the peer review history of their article (what does this mean? ). If published, this will include your full peer review and any attached files.

**Do you want your identity to be public for this peer review?** For information about this choice, including consent withdrawal, please see our Privacy Policy .

Reviewer #1: **Yes: ** Qinbing Fu

Reviewer #2: No

---

## [Editor Report · Acceptance letter]

PONE-D-24-51051R1

PLOS ONE

Dear Dr. Xu,

I'm pleased to inform you that your manuscript has been deemed suitable for publication in PLOS ONE. Congratulations! Your manuscript is now being handed over to our production team.

Kind regards,

on behalf of

Prof Dr Rongchun Han

Academic Editor

PLOS ONE